

# DOC concentrations across a depth-dependent light gradient on a Caribbean coral reef

Benjamin Mueller[1,2,3], Erik H. Meesters[4] and Fleur C. van Duyl[1]

[1] Department of Marine Microbiology and Biogeochemistry, NIOZ Royal Netherlands Institute for Sea Research and Utrecht University, Den Burg, The Netherlands
[2] CARMABI Foundation, Willemstad, Curaçao
[3] Department of Freshwater and Marine Ecology, University of Amsterdam, Amsterdam, The Netherlands
[4] Wageningen Marine Research, Den Helder, The Netherlands

Corresponding author
Benjamin Mueller,
muellerb@ymail.com

## ABSTRACT

Photosynthates released by benthic primary producers (BPP), such as reef algae and scleractinian corals, fuel the dissolved organic carbon (DOC) production on tropical coral reefs. DOC concentrations near BPP have repeatedly been observed to be elevated compared to those in the surrounding water column. As the DOC release of BPP increases with increasing light availability, elevated DOC concentrations near them will, in part, also depend on light availability. Consequently, DOC concentrations are likely to be higher on the shallow, well-lit reef terrace than in deeper sections on the fore reef slope. We measured *in situ* DOC concentrations and light intensity in close proximity to the reef alga *Dictyota* sp. and the scleractinian coral *Orbicella faveolata* along a depth-dependent light gradient from 5 to 20 m depth and compared these to background concentrations in the water column. At 10 m (intermediate light), DOC concentrations near *Dictyota* sp. were elevated by 15 $\mu$mol C L$^{-1}$ compared to background concentrations in the water column, but not at 5 and 20 m (high and low light, respectively), or near *O. faveolata* at any of the tested depths. DOC concentrations did not differ between depths and thereby light environments for any of the tested water types. However, water type and depth appear to jointly affect *in situ* DOC concentrations across the tested depth-dependent light gradient. Corroborative *ex situ* measurements of excitation pressure on photosystem II suggest that photoinhibition in *Dictyota* sp. is likely to occur at light intensities that are commonly present on Curaçaoan coral reefs under high light levels at 5 m depth during midday. Photoinhibition may have thereby reduced the DOC release of *Dictyota* sp. and DOC concentrations in its close proximity. Our results indicate that the occurrence of elevated DOC concentrations did not follow a natural light gradient across depth. Instead, a combination of multiple factors, such as water type, light availability (including the restriction by photoinhibition), and water movement are proposed to interactively determine the DOC concentrations in the close vicinity of BPP.

## INTRODUCTION

Dissolved organic carbon (DOC) is the largest pool of reduced carbon on tropical coral reefs (*Atkinson & Falter, 2003*). A lack of a relationship between particulate organic carbon (POC, as proxy for planktonic primary producers) and DOC concentrations (*Tanaka et al., 2011*), and increased DOC concentrations near the bottom compared to the surface water (*Van Duyl & Gast, 2001*) indicate that benthic primary producers (BPP) are an important source of this DOC. Reef algae and scleractinian corals release a substantial portion of their photosynthetically fixed carbon as DOC into the surrounding water; reef algae generally release more DOC than corals (e.g., *Haas et al., 2011*; *Haas et al., 2013b*). This algal-derived DOC can promote the growth of opportunistic heterotrophic microbes in the water column as well as in the contact zone between corals and algae (*Haas et al., 2013a*; *Haas et al., 2013b*; *Nelson et al., 2013*). Increased microbial respiration in the coral-algal interface causes anoxia (*Gregg et al., 2013*; *Haas et al., 2013a*) in combination with the release of secondary metabolites, and can lead to tissue loss or even coral death (*Barott & Rohwer, 2012*; *Morrow et al., 2013*). Moreover, while most heterotrophic macroorganisms cannot utilize DOC for their nutrition an increasing number of reef sponges are found to predominantly rely on DOC as carbon source (*Yahel et al., 2003*; *De Goeij et al., 2008*; *Mueller et al., 2014a*). As with microbes, sponges also appear to prefer algal- over coral-derived DOC (*Rix et al., 2016*). In the so-called sponge loop, these sponges utilize the energy stored in DOC and make it available to higher trophic levels via subsequent detritus production (*Alexander et al., 2014*; *De Goeij et al., 2013*). Both heterotrophic microbes and DOC-feeding sponges are therefore likely to benefit from elevated DOC concentrations with potential consequences for carbon cycling and overall coral reef functioning (e.g., *Rohwer & Youle, 2010*; *Barott & Rohwer, 2012*; *De Goeij et al., 2013*; *Haas et al., 2016*).

Elevated DOC concentrations in close proximity to BPP have been repeatedly observed on tropical coral reefs (*Van Duyl & Gast, 2001*; *Hauri et al., 2010*; *Mueller et al., 2014b*). However, most studies were conducted in shallow reef areas between 5 and 10 m and little attention was given to deeper reef sections or how DOC concentrations change across depth. Light availability decreases exponentially with depth and is an important environmental parameter that structures benthic communities across the reef slope (e.g., *Bak, 1974*; *Veron, 2000*; *Vermeij & Bak, 2002*). Light availability positively affects the DOC release rates of BPP (*Crossland, 1987*; *Haas et al., 2010b*; *Naumann et al., 2010*; *Barrón, Apostolaki & Duarte, 2014* and references therein). The occurrence of elevated DOC concentrations near BPP were found to be positively correlated with the availability of light (*Mueller et al., 2014b*). We therefore hypothesize that DOC concentrations change with depth and that elevated DOC concentrations near BPP are more likely to occur on the shallow, well-lit reef terrace (5 m) than at the drop off (10 m) or in deeper sections of the fore reef slope (20 m). To test this we measured *in situ* DOC concentrations and light intensity in close proximity to the reef alga *Dictyota* sp. and the scleractinian coral *Orbicella faveolata* (former *Montastraea annularis*) along a depth-dependent light gradient from 5 to 20 m depth and compared these to background concentrations in the water column.

## MATERIALS AND METHODS

Fieldwork was performed under the research permit (#2012/48584) issued by the Curaçaoan Ministry of Health, Environment and Nature (GMN) to the CARMABI foundation.

### Study site and general environmental parameters

Sampling was conducted at Snake Bay (12°8′N, 68°59′W) on the leeward coast of the Island of Curaçao in the Southern Caribbean. The site consists of an approximately 100 m wide sandy reef terrace with patchy coral communities. The reef terrace gradually slopes towards a drop-off that starts at around 10 m depth. The reef then slopes down under a steep angle (20–30°; *Van Duyl (1985)*) and is characterized by a structurally complex reef topography. The hydrodynamics at the study site are mainly dominated by oceanic currents, which generally flows westwards along the island with approx. 50 cm s$^{-1}$ (*Gast et al., 1999*). Due to small tidal differences (10–30 cm), tidal currents are usually neglectable on the narrow Curaçaoan fringing reefs and water currents over the reef terraces are typically around 10–15 cm s$^{-1}$.

Benthic composition was determined from 20 photo quadrats (1 × 1 m) placed at randomized distances and alternatingly on both sides along a 30 m transect line. On March 24, 2017 the benthic cover following the 5, 10, and 20 m isobaths was recorded in the area where the DOC concentrations were quantified before (see DOC concentrations across a depth-dependent light gradient). Percentage cover of most dominant benthic components was quantified from 40 randomly-generated overlaid points on each photograph using Coral Point Count with Excel Extensions (CPCe) (*Kohler & Gill, 2006*). Two photographs of the 5 m transect and five of the 20 m transect had to be excluded from be analysis due to insufficient quality.

In June, 2015 water samples to access bacterial concentrations in the water column 2 m off the reef slope (towards the open ocean) were taken at 15 m depth, following the protocol described in *Dinsdale et al. (2008)* to describe background bacterial abundances. Water samples ($n = 5$) were transported to the nearby CARMABI research station where they were processed and analyzed (for details see Supplemental Information 1).

### DOC concentrations across a depth-dependent light gradient

To quantify DOC concentrations across a depth-dependent light gradient, water samples were taken *in situ* in close proximity (<5 mm) to the reef alga *Dictyota* sp., the scleractinian coral *O. faveolata*, and the water column. Both, *Dictyota* sp. and *O. faveolata* are considered holobionts, including epi- and endophytes and associated microbial communities (*sensu Barott et al., 2011*), jointly affecting the water properties (e.g., DOC concentration) in their close vicinities. At midday on July 24, 2012 between 12:00 hrs and 13:00 hrs (when light intensities are the highest) patches of *Dictyota* sp. and colonies of *O. faveolata* were sampled at 5 (reef flat; high light), 10 (drop-off; intermediate light) and 20 m depth (fore reef slope; low light) (each $n = 5$). In addition, the water column 2 m off the reef slope (towards the open ocean) was sampled ($n = 5$) at the same depths and used to indicate background DOC concentrations (i.e., those not directly affected by DOC release of BPP). Sampling started at 20 m depth and 10 and 5 m were sampled consecutively. At each depth

approx. 10 min were spent to collect all samples. The sampling procedure described by *Van Duyl & Gast (2001)* and modified by *Mueller et al. (2014b)* was followed. In short, water samples were collected using 100 ml acid-washed, polypropylene syringes equipped with a flexible silicon tube attached to their tips. The tube was moved slowly above the surfaces of *Dictyota* sp. and *O. faveolata*, respectively, while collecting water (each $n = 5$). The water column was sampled using a similar syringes ($n = 5$). All water samples were collected facing the water current to avoid potential contamination related to the diver's presence. Ambient light intensity (PAR) was recorded simultaneously while sampling (approx. 10 min; sampling intervals 1 min) using a light meter in a custom-made underwater housing (cosine LI-192SSA underwater quantum sensor connected to LI-1000 data logger; range: PAR 400-700). Water samples were transported (<30 min) to the lab and stored at 4 °C until they were processed later that same day.

## Processing of DOC samples

Water samples collected were filtered (<20 kPa Hg suction pressure) over a 0.2 μm polycarbonate filter (25 mm; Whatman, Maidstone, UK). Prior to filtration, filters, glassware and pipette tips were rinsed three times with acid (10 mL 0.4 M HCl) and twice with sample water (10 mL). Afterwards, 20 mL of sample water was filtered and the filtrate containing DOC was transferred to pre-combusted (4 h at 450 °C) Epa vials (40 mL). Samples were acidified with 6–7 drops of concentrated HCl (38%) to remove inorganic C and stored at 4 °C until analysis. DOC concentrations were measured using the high-temperature catalytic oxidation (HTCO) technique in a total organic C analyzer (TOC-VCPN; Shimadzu, Kyoto, Japan). The instrument was calibrated with a standard addition curve of Potassium Hydrogen Phthalate (0; 25; 50; 100; 200 μmol C L$^{-1}$). Consensus Reference Materials (CRM) provided by DA Hansell and W Chen of the University of Miami (Batch 12; 2012; 41–44 μmol C L$^{-1}$) were used as positive controls for our measurements. Concentrations measured for the batch gave average values (±SD) of $45 \pm 3$ μmol C L$^{-1}$. Average analytical variation of the instrument was <3% (5–7 injections per sample).

## Maximum excitation pressure over photosystem II in *Dictyota* sp.

To explore the occurrence of photoinhibition as a potential explanation for the lack of elevated DOC concentrations observed near *Dictyota* sp. at 5 m depth , an *ex situ* experiment to determine maximum light-dependent reduction of the effective quantum yield of photosystem II ($\Delta F/Fm'$) relative to its maximum at dawn ($Fv/Fm$) (*Iglesias-Prieto et al., 2004*; *Enríquez & Borowitzka, 2010*) was conducted. On March 16, 2017 30 thalli of *Dictyota* sp. were collected on the reef terrace (5 m depth) at Buoy 0 (12° 12′35″N, 68°97′10″) and transported in a dark insulated box to the nearby CARMABI Research Station. The algae were allowed to recover and acclimatize in a flow-through seawater aquarium (flow rate: 7 L min$^{-1}$ to ensure stable temperatures throughout the day) until the commencement of the experiment the following day. On March 17, 2017 at 6:00 hrs (sun rise time: 6:41 hrs; https://www.timeanddate.com/sun/curacao/willemstad?month=3&year=2017) 10 thalli of *Dictyota* sp. were randomly selected from the aquarium and placed in a plastic dish with

sea water. In the lab $Fv/Fm$ was measured in triplicates per thalli with a waterproof PAM fluorometer (Diving PAM, Waltz). After the measurements the thalli were discarded. Light intensity (PAR) and water temperature (°C) were recorded with a light logger (ODYSSEY PAR logger, Dataflow systems; sampling interval 1 min) and a temperature logger (Onset HOBO® Pendant, UA-002-08; sampling intervals 1 min), respectively. After sun set, light intensity gradually increased to approximately 200 µmol photons m$^{-2}$ s$^{-1}$ at 9:00 hrs (see Supplemental Information 1 for light and temperature data). Due to the positioning of the aquarium the light intensity remained stable until 12:00 hrs, when it steeply increased to an average ($\pm$SD) of 1,237 $\pm$ 486 µmol photons m$^{-2}$ s$^{-1}$ for the following three hours. At 15:00 hrs 10 new thalli of *Dictyota* sp. were randomly selected, placed in a plastic dish with sea water, and transferred to the lab. After a dark-adaptation period of 30 min to completely relax photochemical quenching, $\Delta F/Fm'$ was measured in triplicates per thalli.

## Data analysis

Differences in DOC concentrations at the substrate-water-interface of *Dictyota* sp., *O. faveolata* and the water column from 5, 10 and 20 m were tested using one-way ANOVAs, followed by a Tukey HSD post-hoc test in case of significant differences. DOC concentrations were square root transformed to meet assumptions of the analysis. To further explore the combined effects of water types (*Dictyota* sp. and *O. faveolata* contact water, and water column water) and depth, a two-way ANOVA was performed. The maximum excitation pressure over photosystem II ($Q_m$) was calculated as: $Q_m = 1 - [(\Delta F/Fm'$ at noon)/($Fv/Fm$ at dawn)] (*Iglesias-Prieto et al., 2004*). Values that are close to zero indicate that most of the reaction centers of photosystem II remain open, whereas values close to 1.0 indicate that they are closed, suggesting photoinhibition.

## RESULTS

### General environmental parameters

Benthic composition differed between the three sampled depths (Table 1; see Supplemental Information 1 for benthic composition to higher taxonomic levels). While the reef terrace (5 m; high light) was mainly dominated by the non-biological components sand, coral rubble, and bare coral rock, the percentage cover of scleractinian corals, macroalgae, as well as other living taxa increased at the drop off (10 m; intermediate light) and the fore reef slope (20 m; low light). Similarly, with increasing depth the cover of the study organisms *Dictyota* sp. and *O. faveolata* increased from <1% on the reef terrace to 8.8% and 2.6% at the drop off, and 5.5 % and 3.0% on the fore reef slope, respectively. Mean bacterial concentration ($\pm$SD) in the water column at 15 m depth was $9.6 \pm 1.2 \times 10^5$ cells mL$^{-1}$.

### DOC concentrations across a depth-dependent light gradient

At 10 m depth (intermediate light), mean *in situ* DOC concentration in close proximity to the reef alga *Dictyota* sp. was 13 and 11 µmol L$^{-1}$ higher than for the scleractinian coral *O. faveolata* (Tukey HSD, $p = 0.001$) and the water column (Tukey HSD, $p = 0.012$), respectively (Fig. 1 and Supplemental Information 1 for raw data).

In contrast, mean *in situ* DOC concentration at 5 m (high light) and 20 m (low light) did not differ significantly between water types (Table 2A). While mean DOC concentrations

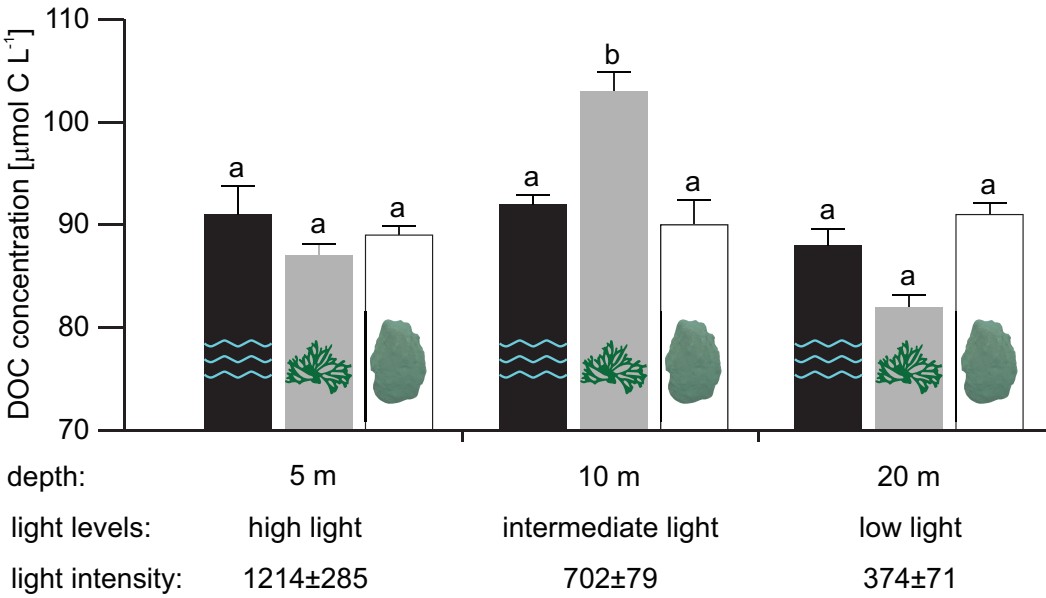

**Figure 1** Mean *in situ* DOC concentrations (*n* = 5, except for water column at 10 m and *Dictyota* sp. at 5 m depth with *n* = 4) measured in the water column (2 m off the reef slope; black) and at the substrate-water interfaces of the reef algae *Dictyota* sp. (dark grey) and the scleractinain coral *Orbicella faveolata* (white) at 5, 10, and 20 m depth. Error bars indicate SE. Different letters indicate differences between water types for each depth (Tukey HSD, $p < 0.05$). Light intensities (mean $\pm$ SD) during the sampling are given in $\mu$mol photons m$^{-2}$ s$^{-1}$.

**Table 1** Community composition (%) of most abundant benthic components at the study site at 5, 10, and 20 m depth. Percentage cover of the studied reef alga *Dictyota* sp. and the scleractinian coral *Orbicella faveolata* are given below.

| Benthic component | Percentage cover (%) | | |
| --- | --- | --- | --- |
| | 5 m | 10 m | 20 m |
| Macroalgae | 0.3 | 14.0 | 31.7 |
| Turf algae | 0.0 | 9.3 | 3.0 |
| Crustose coralline algae | 0.0 | 3.5 | 6.8 |
| Cyanobacteria | 0.0 | 6.6 | 5.3 |
| Scleractinian corals | 1.9 | 24.4 | 32.7 |
| Fire corals | 0.0 | 0.6 | 0.3 |
| Soft corals/gorgonians | 0.0 | 0.3 | 0.3 |
| Sponges | 0.3 | 1.5 | 6.0 |
| Tunicates | 0.0 | 0.1 | 0.0 |
| Sand | 73.5 | 23.1 | 6.3 |
| Bare coral rock | 11.0 | 4.3 | 6.3 |
| Coral rubble | 13.1 | 11.1 | 1.2 |
| **Studied taxa:** | | | |
| *Dictyota* sp. | 0.1 | 8.8 | 5.5 |
| *Orbicella faveolata* | 0.4 | 2.6 | 3.0 |

**Table 2** Results of one-way ANOVAs on the effect of (A) water type (per depth) and (B) depth (per water type) on observed mean *in situ* DOC concentrations. Significant effects are marked with an asterisk.

| | df | F | p | |
|---|---|---|---|---|
| **A. Differences between water types** | | | | |
| **Per depth:** | | | | |
| High light (5 m) | 2 | 0.357 | 0.707 | |
| Intermediate light (10 m) | 2 | 13.378 | 0.001 | * |
| Low light (20 m) | 2 | 0.129 | 0.880 | |
| **B. Differences between depth:** | | | | |
| **Per water type:** | | | | |
| Water column | 2 | 0.148 | 0.864 | |
| *Dictyota* sp. | 2 | 3.682 | 0.06 | |
| *Orbicella faveolata* | 2 | 2.798 | 0.101 | |

**Table 3** Results of a two-way ANOVA on the combined effects of water type and depth on observed mean *in situ* DOC concentrations.

| Factor | df | F | p | |
|---|---|---|---|---|
| Water type | 2 | 4.731 | 0.015 | * |
| Depth | 2 | 0.054 | 0.947 | |
| Water type × Depth | 4 | 3.169 | 0.026 | * |

in close proximity to *Dictyota* sp. at 20 m appear to be depleted relative to concentrations in the water column and near *O. faveolata*, these differences were not found to be significant (Fig. 1). No depth-related differences in mean DOC concentrations were established in any of the tested water types (Table 2B).

However, both water type and depth appear to jointly affect mean *in situ* DOC concentrations along the tested depth-dependent light gradient (significant interaction: water type × depth, $p = 0.026$; Table 3). The sampling depths 5, 10, and 20 m corresponded to a light intensity of $1{,}214 \pm 285$, $702 \pm 79$ and $374 \pm 71$ µmol photons m$^{-2}$ s$^{-1}$ (mean $\pm$ SD), respectively, during the sampling (between 12:00 and 13:00 hrs).

## Maximum excitation pressure over photosystem II in *Dictyota* sp.

At dawn, maximum potential quantum yield ($Fv/Fm$) in *Dictyota* sp. was $0.65 \pm 0.04$ (mean $\pm$ SD), which was reduced by 67% to $0.44 \pm 0.09$ at 15:00 (effective quantum yield; ($\Delta F/Fm'$) (Supplemental Information 1 for raw data). The corresponding $Q_m$ value of 0.33 indicates foremost closed reaction centers of photosystem II and thus suggests the occurrence of photoinhibition at light levels of $1{,}237 \pm 486$ µmol photons m$^{-2}$ s$^{-1}$, which are comparable to those observed *in situ* on the reef terrace at 5 m depth (high light) during midday (see DOC concentrations across a depth-dependent light gradient).

## DISCUSSION

In this study we investigated DOC concentrations in close proximity to the reef alga *Dictyota* sp., the scleractinian coral *O. faveolata*, and in the water column across a depth-dependent light gradient between 5 and 20 m. Due to the positive relationship between light availability and DOC release we hypothesized that DOC concentrations near BPP are highest on the shallow, well-lit reef terrace and decrease along the fore reef slope following a depth-dependent pattern. Elevated DOC concentrations compared to the background concentrations in the water column were only observed near *Dictyota* sp. at an intermediate depths at 10 m, but not at 5 m or 20 m depth, or near *O. faveolata* at any of the tested depth. No depth- and therefore light-related differences in mean DOC concentrations were established in any of the tested water types, however, water type and depth appear to jointly affect *in situ* DOC concentrations across the tested depth-dependent light gradient.

Elevated DOC concentrations in close proximity to BPP occur when DOC release exceeds removal processes. Consequently, environmental parameters, such as light availability (*Haas et al., 2010b*; *Barrón, Apostolaki & Duarte, 2012*), temperature (*Gillooly et al., 2001*; *Haas et al., 2010b*), grazing pressure (*Berman & Holm-Hansen, 1974*), senescence (*Khailov & Burlakova, 1969*), nutrient availability (*Lopéz-Sandoval, Fernandéz & Marañón, 2011*; *Mueller et al., 2016*), and hydrodynamic conditions (*Wild et al., 2012*) affect the DOC release of BPP. In combination with factors which affect the accumulation/removal of DOC near them (e.g., morphology of the BPP, hydrodynamic conditions (*Losee & Wetzel, 1993*; *Escartín & Aubrey, 1995*) and DOC consumption by heterotrophic microbes and sponges (*Gast et al., 1999*; *Yahel et al., 2003*; *Scheffers, Bak & Duyl, 2005*; *De Goeij et al., 2008*; *Haas et al., 2011*; *Nelson et al., 2013*), these parameters interactively determine the DOC concentrations in close vicinity to BPP. The lack of elevated DOC concentrations near *Dictyota* sp. under high light levels at 5 m depth could thus be explained by (1) reduced DOC release, (2) high DOC removal, or (3) a combination of both.

Light availability is generally considered to have a strong positive effect on DOC release of reef algae. However, *Haas et al. (2010b)* reported that this positive correlation in the reef alga *Caulerpa* sp. only held until a maximum light intensity was reached. At these light intensities DOC release rates steeply decreased to levels comparable to those in the dark. They explained this decrease with the onset of photoinhibition at a species-specific light intensity, which is a common phenomenon in coral reef BPP (*Franklin, 1994*; *Hanelt, Li & Nultsch, 1994*; *Brown et al., 1999*; *Hoegh-Guldberg & Jones, 1999*; *Iglesias-Prieto et al., 2004*). While the occurrence of photoinhibition wasn't specifically tested for during the DOC sampling, the corroborative *ex situ* experiment to assess excitation pressure over photosystem II in *Dictyota* sp. suggests that photoinhibition can occur at light levels comparable to those observed *in situ* at 5 m depth on the aforementioned sampling day. Accordingly, photoinhibition may have reduced the DOC release of *Dictyota* sp. on the reef terrace at 5 m depth and therefore contributed to the fact that no elevated DOC concentrations in its close proximity were found.

Similar to light availability, hydrodynamic conditions can affect *in situ* DOC concentrations near BPP in two ways. Either positively, when water movement increases

the metabolism and DOC release rates of BPP by alleviating the limitation of the diffusive boundary layer around them (*Carpenter, Hackney & Adey, 1991*; *Lesser et al., 1994*; *Wild et al., 2012*), or negatively, when water movement and water exchange hamper the accumulation of DOC by dilution (*Hauri et al., 2010*). Water movement generally decreases exponentially as a function of depth (*Shashar et al., 1996*) and significantly higher water movement rates are reported at 5 m compared to 10 or 20 m depth on the reef slope of Curaçao (*Vermeij & Bak, 2003*). Thus, a reduced DOC release rate of *Dictyota* sp. due to photoinhibition in combination with high water movement and water exchange that hamper the accumulation of DOC, could explain the lack of elevated DOC concentrations near *Dictyota* sp. under high light levels at 5 m depth. It can be further assumed that the negative effect of water movement and water exchange on the accumulation of DOC at 10 m was higher than at 20 m, i.e., a higher DOC release rate was necessary to result in elevated DOC concentrations at 10 m. Yet despite higher water movement, elevated DOC concentrations near *Dictyota* sp. were only found at 10, but not at 20 m. One explanation for this difference is that DOC release rates were higher at 10 m (intermediate light) than at 20 m (low light), which is in line with the aforementioned positive relation between light availability and DOC release. Interestingly, whilst not significant, DOC concentrations in close proximity to *Dictyota* sp. at 20 m depth tend to be depleted compared to concentrations near *O. faveolata* or in the water column. Reduced water movement and thus a prolonged water residence time combined with a low, but steady release of bio-available algal DOC (*Nelson et al., 2013*) by *Dictyota* sp., could have stimulated the growth of heterotrophic microbial communities. The bio-available DOC could have further allowed those communities to metabolize otherwise refractory components of the DOC pool and thereby deplete the local DOC stock, as described for the water columns overlying algal-dominated reefs (*Dinsdale et al., 2008*; *Haas et al., 2016*).

No elevated DOC concentrations were observed near the scleractinian coral *O. faveolata* at any of the sampling depths. In general, the DOC release of scleractinian corals is more variable than that of reef algae and an increasing number of studies suggest that scleractinian corals only contribute marginally to the local DOC pool on tropical coral reefs (e.g., *Haas et al., 2010a*; *Naumann et al., 2010*; *Haas et al., 2011*). Furthermore, the massive morphology of *O. faveolata* is less likely to restrict water exchange than the bushy thalli of *Dictyota* sp. and is thereby less favorable for the accumulation of DOC in its vicinity (*Stocking, Rippe & Reidenbach, 2016*). Given the positive effect of light availability on the DOC release by BPP, we expected to find significantly higher DOC concentrations on the shallow and well-lit reef terrace compared to deeper reef sections, following the natural light gradient across depth. Yet no significant differences in the mean DOC concentrations between the sampled depths were observed, neither in close vicinity to the BPP *Dictyota* sp. or *O. faveolata*, nor in the water column. Nevertheless, both water type and depth (and thereby light availability) seem to have interactively determined *in situ* DOC concentrations. This is in accordance with previous findings that suggest that DOC release by, and *in situ* DOC concentrations near BPP are at least partly determined by a substrate-specific relationship with light availability (*Mueller et al., 2014b*). The absence of significant differences in DOC concentrations across the water column was also observed in other studies (*Torréton et*

*al., 1997*; *Nelson et al., 2011*). To date only *Slattery & Lesser (2015)* reported a significant decline in DOC concentration with depth from coral reefs on the Bahamas, albeit this decrease occurred at mesophotic depths below 30 m. This may indicate that at least above mesophotic depths, DOC released by BPP is either quickly mixed and diluted throughout the reef overlying water column and/or taken up by DOC feeding organisms (i.e., heterotrophic bacteria and reef sponges). The abundances of open reef sponges and microbes at our study site, and on Curaçaoan reefs in general, are fairly low (*Gast et al., 1999*; *De Goeij & Van Duyl, 2007*; *Mueller et al., 2014a*; *De Bakker et al., 2017*) compared to abundances at more degraded locations throughout the Caribbean (e.g., *Pawlik et al., 2015* and references therein, *Haas et al., 2016*). However, DOC removal by cryptic sponges living underneath overhangs and in coral cavities, which were not recorded in this study, is estimated to be in the same order of magnitude as gross primary production on these reefs (*De Goeij et al., 2013*).

## CONCLUSION

While light availability has a strong positive effect on the DOC release of BPP, the occurrence of elevated DOC concentrations near them did not follow a natural light gradient across the reef slope in our study system. Instead, a combination of multiple factors, including water type, light availability, which affects the release of DOC (including the restriction by photoinhibition), and water movement, which affects the accumulation/removal of DOC, are proposed to interactively determine the DOC concentrations in the close vicinity of BPP along the reef slope.

## ACKNOWLEDGEMENTS

We thank the staff of Carmabi for their hospitality and logistic support during the field work. We further thank V Chamberland, T Holtrop, Y Mulders, E Van der Ent, R Van der Zande and K Vane for their help during the field work. We are grateful to S Gonzalez for his contribution to the DOC analysis. The manuscript benefitted greatly from comments by M Vermeij on earlier versions of the manuscript.

### Funding

The research leading to these results has received funding from the European Union Seventh Framework Programme (P7/2007-2013) under grant agreement no. 244161 (Future of Reefs in a Changing Environment) and by the Ecology Fund of the Royal Netherlands Academy of Arts and Sciences. The funders had no role in study design, data collection and analysis, decision to publish, or preparation of the manuscript.

### Grant Disclosures

The following grant information was disclosed by the authors:
European Union Seventh Framework Programme (P7/2007-2013): 244161.
Royal Netherlands Academy of Arts and Sciences.

## Competing Interests

The authors declare there are no competing interests.

## Author Contributions

- Benjamin Mueller conceived and designed the experiments, performed the experiments, analyzed the data, wrote the paper, prepared figures and/or tables, reviewed drafts of the paper.
- Erik H. Meesters analyzed the data, reviewed drafts of the paper.
- Fleur C. van Duyl conceived and designed the experiments, analyzed the data, contributed reagents/materials/analysis tools, wrote the paper, reviewed drafts of the paper.

## Field Study Permissions

The following information was supplied relating to field study approvals (i.e., approving body and any reference numbers):

Fieldwork was performed under the research permit (#2012/48584) issued by the Curaçaoan Ministry of Health, Environment and Nature (GMN) to the CARMABI foundation.

## Data Availability

The raw data has been supplied as a Supplementary File.

## Supplemental Information

Supplemental information for this article can be found online at http://dx.doi.org/10.7717/peerj.3456#supplemental-information.

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
