# Peer review of "DOC concentrations across a depth-dependent light gradient on a Caribbean coral reef"

_PeerJ, doi:10.7717/peerj.3456_

## Round 0.1 · original submission · Major Revisions

Both of the Reviewers are experts in the topic of DOC availability on coral reefs, and both expressed significant reservations about this ms., most having to do with the limited data set presented in it. I share the concern that studies such as this one need to be done carefully, with greater replication, and with greater attention to other environmental variables. Note also Reviewer 2's comments about statistical analyses. I recommend that the authors follow the reviewers' advice and expand the replication and context of their study before submitting a revision that addresses the reviewers' concerns.

Reviewer 1 ·

Basic reporting

This manuscript is written nice and concise, the figures are fitting and the raw data is nicely presented in the supplementary information.
One or two sentences could use rephrasing, see comments below

Experimental design

The manuscript is reporting rather on natural history than on experimental data. The sample processing was done thoroughly and I have no concerns here.

Validity of the findings

See below...

Additional comments

The study by Mueller et al. is reporting on the DOC concentrations in close vicinity to different primary producers along a depth gradient in a Caribbean reef. Although the data collection, analysis and statistics are done very well in my opinion I am still not thoroughly convinced that this work is ready to be published.
The authors do a good job presenting their data in the context of the current knowledge on DOC dynamics in coral reef environments and give compelling explanations for the presented patterns.
However, the study is based on 43 DOC values and corresponding PAR measurements only. I would like to see some additional information to support or reject some of the hypotheses
- Oxygen data could be a good indicator for photosynthetic performance and the hypothesized photoinhibition.
- Is there some current data available for these sites?
- The authors mention that sponges might play a significant role in DOC dynamics. Is there data on sponge abundance at the sampling sites?
- The same with microbes, any microbial counts from the sampling sites/depths?
My recommendation here is major revisions. I would however like to recommend reject with the chance of re-submission as I think the authors need to significantly flesh out the story and the conclusions with some more supporting data before publication. I know that it is always hard to add data in retrospect to an existing study, but I am also convinced that it is possible in this case. Curacao is very well studied and to my knowledge is in a rather steady oceanographic setting, so it might be possible to add some information on the general flow conditions, on the study sites, or the overall benthic cover/sponge abundance etc…

Minor comments:
This is an awkward sentence and I think grammatically not correct:
At 10 m depth the distribution of the data of Dictyota sp. differed from that of the water column (Mann-Whitney, p=0.02), with estimated mean DOC concentrations near Dictyota sp. being elevated by 15 μmol L-1 compared to background concentrations.
Please rephrase to something like data distribution of samples collected next to Dictyota…

Figure: keep it uniform, either use icons for everything, or use images for everything. I would just replace the water image with three wavy blue lines…

·

Basic reporting

Good. No comment.

Experimental design

Good. No comment.

Validity of the findings

Good. No comment.

Additional comments

The study by Mueller et al examines concentrations of DOC at three depths on a coral reef in Curacao midday on a single day in midsummer. The authors also measured light (PAR). No other variables are measured for context. The DOC samples were collected from the water as well as near a macroalga (Dictyota) and a coral (O. faveolata). Mean DOC was compared among the three locations at each depth and the Dictyota at the middle depth was higher than other samples, suggesting increased DOC associated with the alga. The manuscript proposes that Dictyota release excess DOC with higher light, but are photoinhibited at the shallowest depth.

The sampling and data in this manuscript appear strong (but see my comments about the inappropriate statistical techniques, which may make this manuscript unpublishable if no statistical differences were observed. The interpretations were excessive, but my edits have easily limited the interpretations as well as suggesting that the Figure 2 be removed (conceptual interpretations are not warranted in a study of such limited scope).

My biggest concern with this manuscript is that the authors simply don't have enough data to say much. While I don't disagree with the methods or interpretations, to publish this and add to the scientific literature would require that they measure more environmental context: flow using ADCPs, nutrients, physiological characteristics of the coral or algae measured, etc. As currently presented this is a very limited dataset and really pushes the idea of a minimum publishable paper.

I recommend minor revision but many of my suggested edits are mandatory for me to recommend acceptance of the paper.

---

## Round 0.2 · accepted · Accept

I believe the authors have properly addressed the comments of the reviewers in preparing their revised manuscript, and have included additional data to address the photo-inhibition issue. Sources and sinks of DOC on coral reefs are currently "hot topics" in marine ecology, and I expect this paper to be part of an interesting debate that furthers our understanding of carbon flux in these important ecosystems.